# LEARNING GRID CELLS BY PREDICTIVE CODING

## ABSTRACT

Grid cells in the medial entorhinal cortex (MEC) of the mammalian brain exhibit a strikingly regular hexagonal firing field over space. These cells are learned after birth and are thought to support spatial navigation but also more abstract computations. Although various computational models, including those based on artificial neural networks, have been proposed to explain the formation of grid cells, the process through which the MEC circuit *learns* to develop grid cells remains unclear. In this study, we argue that predictive coding, a biologically plausible plasticity rule known to learn visual representations, can also train neural networks to develop hexagonal grid representations from spatial inputs. We demonstrate that grid cells emerge robustly through predictive coding in both static and dynamic environments, and we develop an understanding of this grid cell learning capability by analytically comparing predictive coding with existing models. Our work therefore offers a novel and biologically plausible perspective on the learning mechanisms underlying grid cells. Moreover, it extends the predictive coding theory to the hippocampal formation, suggesting a unified learning algorithm for diverse cortical representations.

## 1 INTRODUCTION

Our brain contains a rich set of neural representations of space that help us navigate in an ever-changing world. These include hippocampal place cells (O'Keefe, 1976), which fire when an animal is at a specific spatial position, and grid cells observed in the medial entorhinal cortex (MEC) (Hafting et al., 2005), which fire when an animal occupies multiple positions on a hexagonal or triangular grid. Grid cells have been observed across various species (Fyhn et al., 2008; Yartsev et al., 2011; Doeller et al., 2010), and their remarkable regularity has raised extensive interest in the computational mechanism underlying their emergence. Earlier models have focused on how mechanisms, such as membrane potential oscillation (O'Keefe & Burgess, 2005; Hasselmo et al., 2007) and specialized recurrent connectivity, can generate grid-like firing patterns (Fuhs & Touretzky, 2006; Burak & Fiete, 2009). More recently, research has shown that grid cells can emerge in recurrent neural networks (RNNs) trained using backpropagation through time (BPTT) for path integration tasks. The models are trained to predict their current location by integrating velocity inputs (Cueva & Wei, 2018; Banino et al., 2018; Whittington et al., 2020; Sorscher et al., 2023), providing a normative, task-driven account of the computational problem that the MEC grid cells address. However, the process by which the MEC circuit acquires, or *learns* the grid cells in a biologically plausible way has been largely neglected, despite the fact that grid cells are known to be learned, rather than hardwired at birth (Langston et al., 2010; Wills et al., 2010). Existing learning models (e.g. Weber & Sprekeler (2018)) are highly specialized for grid cells, and it is unclear whether plasticity rules for only one specific cell type exist in the brain.

In this paper, we directly tackle the learning problem underlying the emergence of grid cells using *predictive coding*, an algorithm modeling the plasticity rules for a variety of cortical functions and representations (Rao & Ballard, 1999; Friston, 2005). Our approach to modeling grid cell emergence through predictive coding is motivated by three key factors: Firstly, the predictive coding algorithm can be implemented in predictive coding networks (PCNs) with local computations and Hebbian plasticity (Bogacz, 2017), making it more biologically plausible than learning rules such as back-propagation. Secondly, PCNs have been successful in replicating representations in other regions of the brain, such as the visual cortex (Rao & Ballard, 1999; Olshausen & Field, 1996; Millidge et al., 2024). Thirdly, PCNs have demonstrated the ability to perform hippocampus-related functions, such as associative and sequential memories (Salvatori et al., 2021; Tang et al., 2023; 2024).

The primary contribution of this work is to demonstrate for the first time that grid cells naturally emerge in PCNs trained to represent spatial inputs with biologically plausible plasticity rules. In this work we:

- show that hexagonal grid cells develop as the latent representations of place cells in classical PCNs (Rao & Ballard, 1999; Olshausen & Field, 1996) with sparse and non-negative constraints;

- train a dynamical extension of classical PCNs, called temporal predictive coding network (tPCN) (Millidge et al., 2024), in path integration tasks and observe that the latent activities of the tPCN develop hexagonal, grid-like representations, similar to what has been discovered in RNNs;

- develop an understanding of grid cell emergence in tPCN, by showing analytically that the Hebbian learning rule of tPCN implicitly approximates truncated BPTT (Williams & Peng, 1990);

- show that tPCN can robustly develop grid cells under different architectural choices, and even without velocity inputs in path integration.

Overall, our results present an effective and plausible learning rule for hexagonal grid cells in the MEC based on predictive coding. We offer a novel extension of predictive coding theory, which has traditionally been used to model visual representations (Rao & Ballard, 1999; Olshausen & Field, 1996), to encompass spatial representations in the MEC. Our findings therefore offer a novel understanding of how a single, unified learning algorithm can be employed by different brain regions to represent inputs of various levels of abstraction.

## 2 RELATED WORK

**Computational Models of Grid Cells**   The periodicity of grid cells inspired early models of grid cells based on membrane potential oscillations, where the periodic firing of grid cells results naturally from the interference between somatic and dendritic oscillators in MEC pyramidal neurons (O'Keefe & Burgess, 2005; Hasselmo et al., 2007). These models were subsequently extended to incorporate multiple networks of oscillatory neurons (Zilli & Hasselmo, 2010). However, these models lack biological plausibility as they require an unrealistically large number of networks (Giocomo et al., 2011). Another major family of models leverages the recurrent attractor networks and obtains grid firing patterns (Fuhs & Touretzky, 2006; Burak & Fiete, 2009; Ocko et al., 2018) by hand-tuning the recurrent connectivity to form a center-surround structure. These networks perform robust and accurate path integration (Burak & Fiete, 2009) and can explain experimental observations such as the deformation of grid cells in irregular environments (Ocko et al., 2018). However, as pointed out by Sorscher et al. (2023), these models lack an explanation for the underlying spatial task that gives rise to the specific recurrent connectivity.

To address this gap, recent studies have explored the question *'If grid cell is the answer, what is the question?'*. Dordek et al. (2016) showed that grid cells emerge as the non-negative principal components of place cells, while Stachenfeld et al. (2017) proposed that grid cells form a basis for predicting future observations. Other studies have focused on the multi-modularity of grid cells by optimizing biologically constrained objective functions (Dorrell et al., 2022; Schaeffer et al., 2024). Notably, multiple research tracks have found that RNNs trained to perform path integration tasks will develop hexagonal grid representations in their latent states (Cueva & Wei, 2018; Banino et al., 2018; Whittington et al., 2020), suggesting that grid cells emerge as a result of successful navigation. These findings were further reinforced by Sorscher et al. (2023), who analytically demonstrated that path integration with certain implementation choices, such as non-negativity, is a sufficient condition for the emergence of grid cells, clarifying earlier controversies (Schaeffer et al., 2022). However, none of these works have addressed how the MEC/hippocampal network learns the grid cells. The RNN models are trained by BPTT, a learning rule unlikely to be employed by the brain (Lillicrap & Santoro, 2019). Even though the principal component model by Dordek et al. (2016) can be learned with the plausible Sanger's rule (Sanger, 1989), it has been shown that principal component analysis (PCA) cannot be applied to other brain regions such as the visual areas (Olshausen & Field, 1996), and Sanger's rule cannot be generalized to dynamical tasks such as path integration. Earlier models of the learning process of grid cells have explored plausible learning rules such as spike time-dependent plasticity (Widloski & Fiete, 2014) and variants of Hebbian learning rules (Kropff & Treves, 2008) within networks of excitatory and inhibitory neurons (Weber & Sprekeler, 2018). However, these learning rules are highly specialized, and have not been shown to reproduce

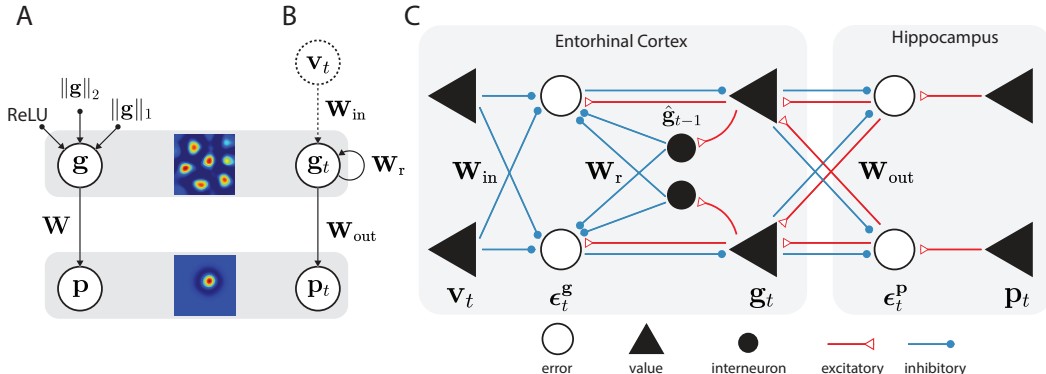

Figure 1: **Architecture and circuit implementation of PCNs.** A: Sparse, non-negative PCN as a generative model. During learning, $\mathbf{p}$ is given and the latent $\mathbf{g}$ and $\mathbf{W}$ are inferred and learned through a type of EM algorithm. B: Simlar to A, but with dynamic inputs $\mathbf{p}_t$ and recurrent weights $\mathbf{W}_r$. The dashed velocity inputs are optional (see Section 4.4). C: Circuit implementation of tPCN, adapted from Tang et al. (2024) with a mapping to MEC and hippocampus.

representations from other brain regions with non-spatial tasks. Recent works have also modeled the hippocampal formation using generative models with plausible learning rules similar to predictive coding (George et al., 2024; Bredenberg et al., 2021), though these studies did not address 2D spatial learning.

**Predictive Coding**   Predictive coding has been an influential theory in understanding cortical computations (Friston, 2005; Rao & Ballard, 1999; Bogacz, 2017) and has been applied to modeling various cortical functions (see Millidge et al. (2021) for a review). Specifically, in the visual cortex, PCNs develop realistic visual representations such as Gabor-like receptive fields in response to both static (Rao & Ballard, 1999; Olshausen & Field, 1996) and moving stimuli (Millidge et al., 2024). Recently, theories have been developed to describe hippocampo-neocortical interactions using predictive coding (Barron et al., 2020), and PCNs have demonstrated the ability to memorize and retrieve static and dynamic visual patterns, a key function of the hippocampus (Salvatori et al., 2021; Tang et al., 2023; 2024). Our work explores whether the representational learning capabilities of predictive coding can be extended to the hippocampal formation, which has so far only been functionally modeled by PCNs.

The computations of PCNs use only local neural dynamics and Hebbian plasticity, making it biologically more plausible than backpropagation (Whittington & Bogacz, 2017). It has also been shown that predictive coding approximates backpropagation both in theory and practice (Whittington & Bogacz, 2017; Song et al., 2024; Pinchetti et al., 2024). Unlike many other Hebbian learning rules, predictive coding can be extended to temporal predictive coding networks (tPCNs), which use recurrent connections to process dynamic stimuli (Millidge et al., 2024). However, while Millidge et al. (2024) demonstrated that tPCNs approximate Kalman filtering, the relationships between tPCNs and RNNs remain unclear. In this work, we train tPCNs for path integration and compare their performance with RNNs both analytically and experimentally in this context.

## 3 MODELS

**Non-negative Sparse PCN**   We first investigate the classical PCN (Rao & Ballard, 1999) for its ability to form grid representations. Assuming a place cell input $\mathbf{p} \in \mathbb{R}^{N_p}$ that represents a location in 2D space as an $N_p$-dimensional vector, a simple 2-layer PCN generates predictions of $\mathbf{p}$ using its latent activities $\mathbf{g} \in \mathbb{R}^{N_g}$ (which will develop grid-like representations) and a weight matrix $\mathbf{W}$ (Fig 1A). The generative model minimizes the following loss function subject to two constraints:

$$\mathcal{L}_{\text{PCN}} = \|\mathbf{p} - \mathbf{W}\mathbf{g}\|_2^2 + \|\mathbf{g}\|_2^2 + 2\lambda\|\mathbf{g}\|_1 \tag{1}$$

where $\|\mathbf{g}\|_2^2$ constrains the $l2$ norm of the latent $\mathbf{g}$ and $\lambda\|\mathbf{g}\|_1$ enforces sparsity, similar to the sparse coding model (Olshausen & Field, 1996). This loss function is minimized via an expectation-

maximization (EM) algorithm, alternating between the optimization over $\mathbf{g}$ (inference) and $\mathbf{W}$ (learning) (see Appendix A.1 for the training algorithm):

$$\Delta\mathbf{g} \propto -\nabla_{\mathbf{g}}\mathcal{L}_{\text{PCN}} = -\mathbf{g} - \lambda\text{sgn}(\mathbf{g}) + \mathbf{W}^{\top}\boldsymbol{\epsilon}^{\mathbf{P}}; \quad \mathbf{g} \leftarrow \text{ReLU}(\mathbf{g} + \Delta\mathbf{g}) \tag{2}$$

$$\Delta\mathbf{W} \propto -\nabla_{\mathbf{W}}\mathcal{L}_{\text{PCN}} = \boldsymbol{\epsilon}^{\mathbf{P}}\mathbf{g}^{\top} \tag{3}$$

where $\boldsymbol{\epsilon}^{\mathbf{P}} := \mathbf{p} - \mathbf{W}\mathbf{g}$ and we apply a ReLU to the inference dynamics to constrain the latent activities to be non-negative. The inference and learning dynamics can be implemented in a plausible circuit (Bogacz, 2017). After convergence, we examine the firing fields of the latent activities $\mathbf{g}$.

**Path Integrating tPCN**    To account for the learning of spatial representations in moving animals, we also investigate tPCN that extends the classical PCNs to the temporal domain (Millidge et al., 2024; Tang et al., 2024) in path integration tasks (Fig. 1B). The model receives a series of place cell activities $\mathbf{p}_1, ..., \mathbf{p}_T$ and velocity inputs $\mathbf{v}_1, ..., \mathbf{v}_T$ that represent the trajectory of an agent moving in a 2D space, and minimizes the following loss function at each time step $t$:

$$\mathcal{L}_{\text{tPCN},t} = \|\mathbf{p}_t - f(\mathbf{W}_{\text{out}}\mathbf{g}_t)\|_2^2 + \|\mathbf{g}_t - h(\mathbf{W}_{\text{r}}\hat{\mathbf{g}}_{t-1} + \mathbf{W}_{\text{in}}\mathbf{v}_t)\|_2^2 \tag{4}$$

where $f$ and $h$ are both nonlinear activation functions, and $\mathbf{W}_{\text{in}}$, $\mathbf{W}_{\text{r}}$ and $\mathbf{W}_{\text{out}}$ are weight matrices projecting the predictions. We define $\boldsymbol{\epsilon}_t^{\mathbf{P}} := \mathbf{p}_t - f(\mathbf{W}_{\text{out}}\mathbf{g}_t)$, $\boldsymbol{\epsilon}_t^{\mathbf{g}} := \mathbf{g}_t - h(\mathbf{W}_{\text{r}}\hat{\mathbf{g}}_{t-1} + \mathbf{W}_{\text{in}}\mathbf{v}_t)$. The model learns by first optimizing the loss function with respect to $\mathbf{g}_t$ via gradient descent:

$$\Delta\mathbf{g}_t \propto -\nabla_{\mathbf{g}_t}\mathcal{L}_{\text{tPCN},t} = -\boldsymbol{\epsilon}_t^{\mathbf{g}} + \mathbf{W}_{\text{out}}^{\top}f'(\mathbf{W}_{\text{out}}\mathbf{g}_t)\boldsymbol{\epsilon}_t^{\mathbf{P}} \tag{5}$$

and then optimizing weights by:

$$\{\Delta\mathbf{W}_{\text{out}}, \Delta\mathbf{W}_{\text{r}}, \Delta\mathbf{W}_{\text{in}}\} \propto -\nabla_{\{\mathbf{W}_{\text{out}},\mathbf{W}_{\text{r}},\mathbf{W}_{\text{in}}\}}\mathcal{L}_{\text{tPCN},t}$$
$$= \{f'(\mathbf{W}_{\text{out}}\mathbf{g}_t)\boldsymbol{\epsilon}_t^{\mathbf{P}}\mathbf{g}_t^{\top}, h'(\tilde{\mathbf{g}}_t)\boldsymbol{\epsilon}_t^{\mathbf{g}}\hat{\mathbf{g}}_{t-1}^{\top}, h'(\tilde{\mathbf{g}}_t)\boldsymbol{\epsilon}_t^{\mathbf{g}}\mathbf{v}_t^{\top}\} \tag{6}$$

where $f'$ and $h'$ are Jacobians of the nonlinear functions $f$ and $h$, and $\tilde{\mathbf{g}}_t := \mathbf{W}_{\text{r}}\hat{\mathbf{g}}_{t-1} + \mathbf{W}_{\text{in}}\mathbf{v}_t$. After the inference (Equation 5) converges, we set $\hat{\mathbf{g}}_t$ to the converged value of $\mathbf{g}_t$, which will be used for optimizing the objective function at the next time step i.e., $\mathcal{L}_{\text{tPCN},t+1}$. The model is trained on a large number of trajectories $\{\mathbf{v}_t, \mathbf{p}_t\}$ and after training, a set of velocity inputs from unseen trajectories is presented to the model. The model then performs a forward pass through time and layers to predict the positions encoded by place cells (see Appendix A.1 for the training and testing algorithms of tPCN):

$$\mathbf{g}_t = h(\mathbf{W}_{\text{r}}\mathbf{g}_{t-1} + \mathbf{W}_{\text{in}}\mathbf{v}_t), \quad \hat{\mathbf{p}}_t = f(\mathbf{W}_{\text{out}}\mathbf{g}_t) \tag{7}$$

The model is evaluated on 1) the accuracy of path integration position prediction $\hat{\mathbf{p}}_t$ and 2) the firing fields of the latent $\mathbf{g}$. When both $f$ and $h$ are linear, these computations can be plausibly implemented in a neural circuit shown in Figure 1C, with local inference computations (Equation 5) and Hebbian learning rules (Equation 6) (Millidge et al., 2024). When the activation functions involve only local nonlinearity, such as tanh or ReLU, the Jacobians are diagonal and the inference and learning rules remain local and Hebbian (Millidge et al., 2022), and additional circuitry components can be included to plausibly implement the nonlinearities (Whittington & Bogacz, 2017). Within the context of spatial representation learning, this circuit implementation can be naturally mapped to the circuitry of the hippocampal formation. We discuss the relationship of this circuit implementation to existing and potential experimental evidence in the Discussion section.

**Input of the Model**    In models discussed in this work, we assume that grid cells are inferred as latent representations of place cells. Although previous models have followed the opposite direction of the relationship, several strands of experimental evidence have suggested the emergence of grid cells as a result of place cells, including the earlier development of place cells (Bush et al., 2014; Langston et al., 2010; Wills et al., 2010). In both PCN and tPCN models, the place cell inputs are constructed as 2D difference-of-softmaxed-Gaussian (DoS) curves flattened into 1D vectors, which have been shown to yield hexagonal grid representations in RNNs (Schaeffer et al., 2022; Sorscher et al., 2023). The firing centers of the place cells are uniformly distributed across a 2D environment. For PCN, the inputs are $N_x$ evenly distributed locations in the environment ($N_x$ large enough to cover the whole environment) represented by the $N_p$ place cells. For tPCN, the trajectories for the path integration task are obtained by simulating an agent performing a smooth random walk in the square environment. At each point in time, the $N_p$ place cells will be uniquely activated,

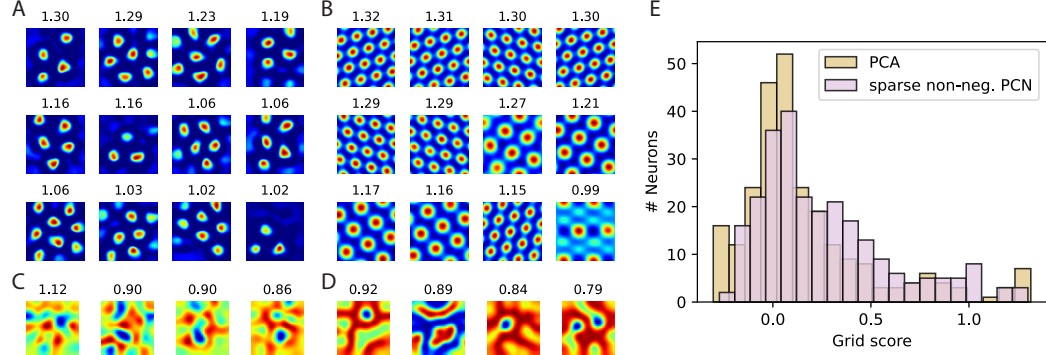

Figure 2: **Grid cells developed in PCN.** A: Latent representations of a sparse, non-negative PCN, resembling hexagonal grid cells in the MEC. Numbers in the title reflect the grid scores. B: Grid cells obtained via the pattern formation theory/non-negative PCA discussed in Sorscher et al. (2023); Dordek et al. (2016). C, D: Latent representations without sparsity or non-negativity, respectively. E: Distribution of grid scores of the representations in A and B.

representing the agent's current location. The velocity inputs $\mathbf{v}_t$ are 2D vectors representing the speed of the simulated agent on the $x$ and $y$ coordinates at time step $t$. The effect of boundaries is simulated by slowing down the agent and reverting its moving direction near the borders of the environment. We sample a large number of trajectories to cover the whole simulated environment for training.

## 4 RESULTS

### 4.1 SPARSE NON-NEGATIVE PCN DEVELOPS LATENT GRID CELLS

Here we examine whether the sparse non-negative PCN can develop hexagonal, grid-like latent representations of the space after training, by plotting each latent neuron's responses to the $N_x = 900$ locations in the 2D space. We use $N_p = 512$ and $N_g = 256$. The "gridness" of the 2D latent representations is evaluated using the *grid score* metric, commonly employed in both experimental and computational studies (Sargolini et al., 2006; Banino et al., 2018) (see A.3 for grid score calculation). We found that this simple, 2-layer PCN can develop hexagonal grid cells similar to those observed in the MEC (Figure 2A). For comparison, we reproduce the results from Dordek et al. (2016) and Sorscher et al. (2023), which show theoretically that performing non-negative PCA on the place cell inputs is guaranteed to produce hexagonal grid representations as the principal components of the $N_x \times N_p$ place cell input matrix. The visual results of the reproduction are shown in Figure 2B, and we compare the distribution of grid scores of the PCN's latent neuron firing fields with those of the non-negative principal components in Figure 2E. The grid scores between our sparse non-negative PCN and non-negative PCA are similarly distributed.

Why does the sparse, non-negative PCN develop hexagonal grid cells? While a precise analytical explanation is left for future work, we offer an intuitive hypothesis here. When presented with a batch $N_x$ of place cell inputs, the objective of PCN (Equation 1) can be written compactly as:

$$\mathcal{L}_{\text{PCN}} = \|\mathbf{P} - \mathbf{G}\mathbf{W}^\top\|_F^2 + \sum_{i=1}^{N_x} \|\mathbf{g}_i\|_2^2 + 2\lambda\|\mathbf{g}_i\|_1 \tag{8}$$

where $\mathbf{P} \in \mathbb{R}^{N_x \times N_p}$ is the place cell activities across $N_x$ locations, and $\mathbf{G} \in \mathbb{R}^{N_x \times N_g}$ represents grid cell responses. On the other hand, the objective function of PCA is:

$$\mathcal{L}_{\text{PCA}} = \|\mathbf{P} - \mathbf{G}\mathbf{M}\|_F^2 \quad \text{s.t.} \quad \mathbf{G}\mathbf{G}^\top = \mathbf{I}_{N_x} \tag{9}$$

where $\mathbf{M}$ is the $N_g \times N_p$ readout matrix. The constraint $\mathbf{G}\mathbf{G}^\top = \mathbf{I}_{N_x}$ in Equation 9 enforces orthonormality of the grid cell matrix $\mathbf{G}$ columns, meaning they are orthogonal and have unit norm. We hypothesize that the constraint $\|\mathbf{g}_i\|_2^2 + 2\lambda\|\mathbf{g}_i\|_1$ for our sparse PCN achieves this orthonormality implicitly: while the constraints are imposed on the rows of $\mathbf{G}$, the overall sparsity of entries

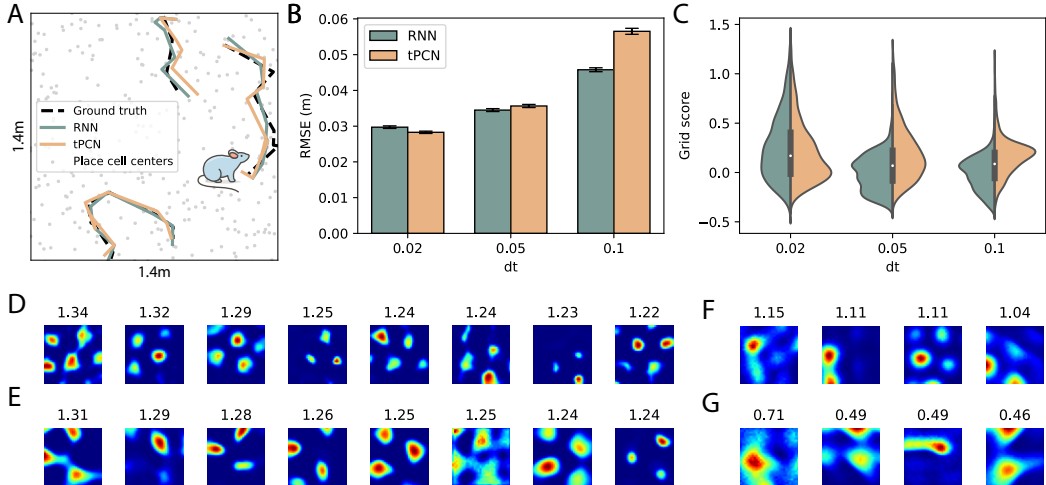

Figure 3: **tPCN in path integration**. A: Visual demonstration of the performance of tPCN and RNN in path integration. B: RMSEs between the decoded and ground-truth 2D positions by tPCN and RNN with different agent moving speed. C: Grid score distributions of tPCN and RNN with different agent moving speed. D, E: Firing fields of latent neurons in a tPCN and an RNN respectively, when $dt = 0.02$. F, G: Firing fields of latent neurons in a tPCN and an RNN respectively, when $dt = 0.1$.

in **G** could induce orthogonality among its columns, with the $l_2$ term constraining the norm of the columns to achieve normality implicitly. Indeed, Figure 2C shows that if we remove the sparsity constraint, the latent neurons' firing fields will no longer be hexagonal. Similarly, without `ReLU` i.e., non-negativity applied to the inference dynamics, we also could not obtain hexagonal grid cells (Figure 2D). It is worth noting that although (non-negative) PCA can be learned with the biologically plausible Sanger's rule (Sanger, 1989), it lacks PCN's generalizability to different architectures (Salvatori et al., 2022) and to other brain regions such as the visual cortex (Olshausen & Field, 1996; Rao & Ballard, 1999). However, it can be noticed that the grid cells by PCN lack the multi-modularity of the grid cells by non-negative PCA i.e., grid cells with different firing periods. We suspect that although sparse PCNs can approximate the orthonormality of latent variables, they lack PCA's ability to extract latent variables ordered by the amount of explained variance in data, with higher variance naturally corresponding to larger spatial scales and vice versa.

### 4.2 TPCN DEVELOPS GRID CELLS BY PATH INTEGRATION

Although training a static PCN with a large number of place cell activations can already give rise to brain-like hexagonal grid cells, the emergence of grid cells is known to rely on dynamic motion of animals (McNaughton et al., 2006; Winter et al., 2015). Therefore, we investigate tPCN in a path integration task, where the simulated agent uses dynamic velocity inputs to determine its current position. As a reference, we compare tPCN with RNNs trained in path integration, which have been shown to develop hexagonal grid cells (Cueva & Wei, 2018; Banino et al., 2018; Sorscher et al., 2023) and share the same graphical structure as tPCN (Figure 1B). However, it is important to note that RNNs are trained with the biologically implausible backpropagation-though-time (BPTT) algorithm, which requires "unrolling" of the network through time, a process unlikely to occur in the brain (Lillicrap & Santoro, 2019).

We first evaluate whether tPCN can learn to perform the path integration task using local and Hebbian learning rules. We trained a tPCN model with $N_g = 2048$ latent neurons on trajectories within a 1.4m × 1.4m environment represented by $N_p = 512$ place cells. After training, we tested the model on a set of unseen trajectories with velocity input $\mathbf{v}_t$, and assessed whether the tPCN and RNN models could predict the correct positions using Equation 7. As the output of the networks is the $N_p$-dimensional population activity of the place cells, we calculate the predicted 2D positions by averaging the center positions of the 3 most active place cells in the output $\hat{\mathbf{p}}_t$, and calculate the root mean square error (RMSE) between the decoded and ground-truth 2D positions. The visual and numerical results are shown in Figure 3A and B, where we also varied a scaling factor $dt$ of the

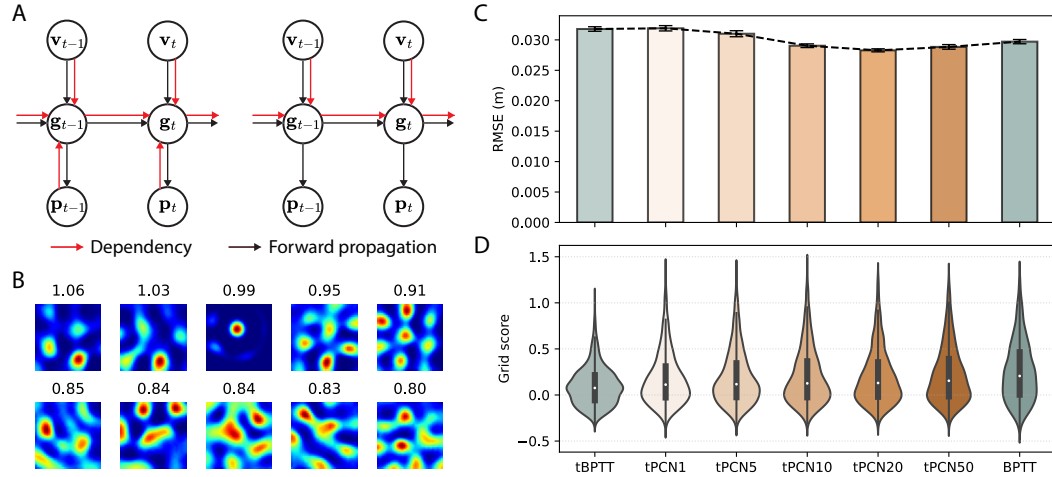

Figure 4: **Comparing tPCN and tBPTT.** A: Dependencies of latent grid cells in tPCN and RNN trained with 1-step tBPTT. Black arrows indicate the flow of computations during a forward pass and red arrows indicate the dependency of latent variables. B: Firing fields of the latent neurons of an RNN trained by 1-step tBPTT. C, D: Path integration RMSE and grid score distributions of 1-step tBPTT, BPTT and tPCNs with different inference iterations. "tPCNk" indicates tPCN trained with k inference iterations.

simulated agent's speed, sampled from a Rayleigh distribution with mean 1, to test the robustness of the results. Note that we do not intend to model physiologically realistic speed of animals with these values. The performance of tPCN is comparable to that of the RNN, though it slightly deteriorates when the agent moves at higher speeds.

Next, we examine whether the tPCN model develops grid-like representations in its latent layer during path integration. We plot the firing fields of the 2048 latent neurons given an unseen set of trajectories covering the entire space. The neurons with the highest grid scores are shown in Figure 3C, which reveals a grid-like, hexagonal firing pattern with high grid scores. Visually, these grid cells are similar to those in a trained RNN with the same architecture shown in Figure 3E, replicating the results from (Sorscher et al., 2023). To systematically compare the grid cells in tPCN and RNN, we plot the distribution of grid scores in both models as a function of the movement speed of the agent in the environment in Figure 3C. When the movement is slow, the grid score distributions are similar between tPCN and RNN. However, as the $dt$ increases to 0.05 and 0.1, tPCN tends to have higher grid scores than RNN. This is visually reflected in Figure 3F (tPCN) and G (RNN), which shows the latent representations developed by tPCN largely retain the grid-like pattern whereas firing centers of many of the RNN neurons no longer form a grid when $dt = 0.1$. Interestingly, the band-like representations present in both models in this case are observed in MEC (Krupic et al., 2012), although their existence is controversial (Navratilova et al., 2016).

### 4.3 TPCN APPROXIMATES TRUNCATED BPTT

Next, we asked why hexagonal grid representations emerge both when training a tPCN using a BPTT-free Hebbian learning rule and when training an RNN using BPTT. We provide an analytical comparison between the learning rules of tPCN and RNN. Assuming a vanilla, sequence-to-sequence RNN with exactly the same graphical structure as in Figure 1A, its dynamics can be recursively described as:

$$\mathbf{g}_t = h(\mathbf{W}_r \mathbf{g}_{t-1} + \mathbf{W}_{in} \mathbf{v}_t); \quad \hat{\mathbf{p}}_t = f(\mathbf{W}_{out} \mathbf{g}_t) \tag{10}$$

The loss that this RNN is trained to minimize is the cumulative prediction error:

$$\mathcal{L}_{RNN} = \sum_{t=1}^{T} \mathcal{L}_{RNN,t} = \sum_{t=1}^{T} \|\mathbf{p}_t - \hat{\mathbf{p}}_t\|_2^2 \tag{11}$$

Suppose BPTT is performed at every step $t$ to update weights in this RNN, the learning rule for $\mathbf{W}_r$ at step $t$ can be expressed as (see Appendix A.2 for derivations):

$$\Delta\mathbf{W}_r^{\text{RNN}} = \sum_{k=1}^{t} \frac{\partial \mathbf{g}_t}{\partial \mathbf{g}_k} h'(\tilde{\mathbf{g}}_t)\mathbf{W}_{\text{out}}^{\top} f'(\mathbf{W}_{\text{out}}\mathbf{g}_t)\boldsymbol{\epsilon}_t^{\mathbf{P}} \underline{\mathbf{g}_{k-1}^{\top}} \tag{12}$$

where $\boldsymbol{\epsilon}_t^{\mathbf{P}}$ denotes the prediction error $\mathbf{p}_t - \hat{\mathbf{p}}_t$ and the $\frac{\partial \mathbf{g}_t}{\partial \mathbf{g}_k}$ terms correspond to the unrolling in BPTT, which can be factorized into a chain of partial derivatives (Bellec et al., 2020). On the other hand, for tPCN, if we assume that the inference dynamics in Equation 5 has fully converged ($\Delta\mathbf{g}_t = 0$) at the time of weight update, the learning rule of tPCN can be written as (see Appendix A.2 for derivations):

$$\Delta\mathbf{W}_r^{\text{tPCN}} = h'(\tilde{\mathbf{g}}_t)\mathbf{W}_{\text{out}}^{\top} f'(\mathbf{W}_{\text{out}}\mathbf{g}_t)\boldsymbol{\epsilon}_t^{\mathbf{P}} \underline{\hat{\mathbf{g}}_{t-1}^{\top}} \tag{13}$$

Two key differences between these learning rules stand out. First, tPCN does not involve the recursive unrolling term, thereby avoiding the need to maintain a perfect memory of all preceding hidden states. Second, instead of using the forward-propagated $\mathbf{g}_{t-1}$ as in Equation 10, tPCN employs the inferred $\hat{\mathbf{g}}_{t-1}$ from Equation 5 (underlined). The first difference suggests an equivalence between tPCN and RNN trained with truncated BPTT (tBPTT) with a truncation window of size 1 (1-step tBPTT) (Williams & Peng, 1990), where the RNN does not backpropagate *any* hidden states through time when updating the weights. This characteristic could potentially harm the RNN's performance as it cannot effectively perform temporal credit assignment. However, the second difference partially solves this problem, as $\hat{\mathbf{g}}_{t-1}$ is inferred following Equation 5, which includes the term $\boldsymbol{\epsilon}_{t-1}^{\mathbf{P}}$ that communicates the place cell prediction error at step $t-1$. Therefore, when $\mathbf{W}_r$ is updated at step $t$, the $\hat{\mathbf{g}}_{t-1}^{\top}$ term in $\Delta\mathbf{W}_r^{\text{tPCN}}$ will effectively form an eligibility trace (Bellec et al., 2020) that allows the model to access historical prediction errors on the place cell level. Figure 4A illustrates this difference between tPCN and RNN trained by 1-step tBPTT, highlighting the dependency of tPCN hidden states on past place cell activations. In Appendix A.2 we also discuss the relationship between the update rules for $\mathbf{W}_{\text{in}}$ and $\mathbf{W}_{\text{out}}$ in these two models.

To verify this theoretical difference, we compare tPCN with RNNs trained by tBPTT in the path integration task. Since $\hat{\mathbf{g}}_t$ in tPCN is initialized by a forward pass $f(\mathbf{W}_r\hat{\mathbf{g}}_{t-1})$ and then updated by the iterative inference (Appendix A.1), the behavior of 1-step tBPTT, which computes its latent states via a forward pass at each time step, should be closer to tPCN with fewer inference iterations. Therefore, we evaluate tPCN with various inference iterations. Figure 4B shows the grid cells learned by an RNN trained with 1-step tBPTT, which still exhibit hexagonal grid firing fields, though with lower grid scores than those from full BPTT. This suggests that backpropagating the error through all time steps is not entirely necessary for RNNs to generate grid cell-like representations. In Figure 4C we show the path integration performance of RNN by 1-step tBPTT and BPTT, as well as tPCNs with different inference iterations from 1 to 50. As can be seen, tPCN with a single inference iteration has identical performance to RNN trained by tBPTT, and its performance will improve as we increase the number of inference iterations but will saturate around 20 iterations. Overall, this graph suggests that tPCN with 5 or more inference iterations can effectively perform temporal credit assignment that improves upon tPCN1 or 1-step tBPTT, potentially due to the eligibility trace. However, this eligibility trace arises from local inference dynamics (Equation 5) rather than from unrolling the RNN graph as in Bellec et al. (2020). This improvement is also reflected in the grid scores (Figure 4D), although increasing the inference iterations does not necessarily result in better grid score representations. We suspect that although the gridness of latent representations is somewhat related to path integration performance, their relationship is not linear. It is also worth noting that to fully evaluate the similarities and differences between BPTT and tPCN, an in-depth comparison is needed across different tasks and versions of tBPTT. We aim to investigate this question in future works as it is beyond the scope of this paper.

### 4.4 ROBUSTNESS OF GRID CELL REPRESENTATIONS IN TPCN

Inspired by Schaeffer et al. (2022), we examine the robustness of our results against different architectural choices of the tPCN model, to understand what contributes to the emergence of grid cells within tPCN. Specifically, we vary the following components of the model: 1) Encoding of the place cell activities; 2) Output nonlinearity $f$; 3) Recurrent nonlinearity $h$; 4) Environment sizes; 5) Latent sizes and 6) Velocity input to the model. The baseline model has DoS place cell encodings, $h = \texttt{ReLU}$, $f = \texttt{softmax}$, 1.4m $\times$ 1.4m environment and latent size 2048 with velocity inputs.

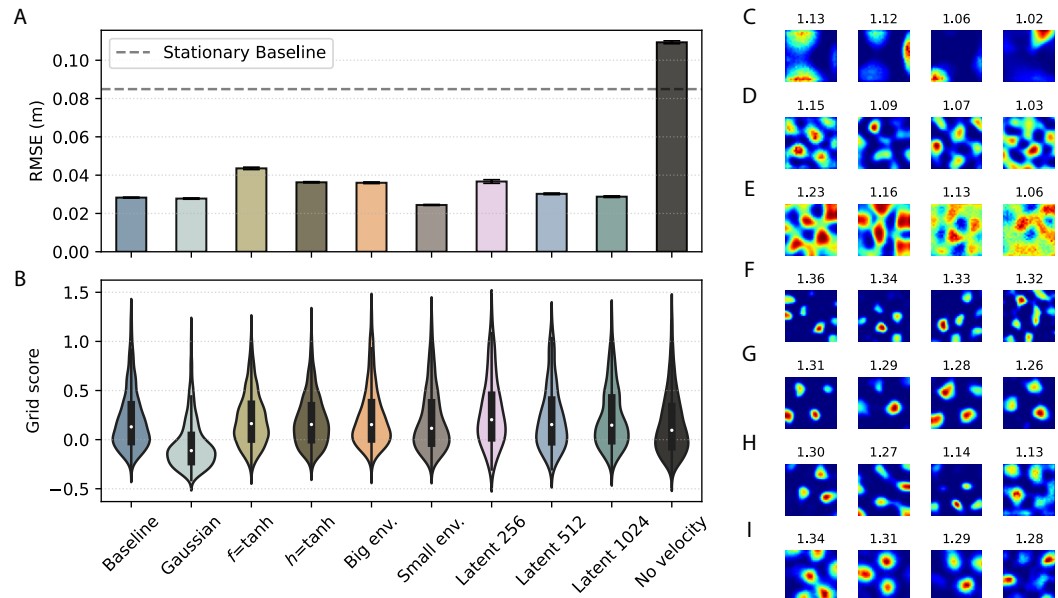

Figure 5: **Robust emergence of grid cells in tPCN.** A, B: Path integration RMSE and grid scores of tPCN in different setups. "Stationary baseline" refers to a model that always predicts the initial position regardless of movement. C-I: Firing fields of latent neurons in tPCNs with C: Gaussian place cells; D: $f =$tanh; E: $h =$tanh; F: 1.8m × 1.8m environment; G: 1.2m × 1.2m environment; H: 256 latent neurons; I: tPCN without velocity input.

We first examine whether replacing the place encoding with Gaussian curves affects the model's performance. As shown in Figure 5A, B and C, the Gaussian place cells do not affect the path integration performance, but the latent representations are no longer hexagonal. This is consistent with earlier findings that the DoS place cell encoding is necessary for hexagonal grid cells (Dordek et al., 2016; Sorscher et al., 2023; Schaeffer et al., 2022).

The choices of $f$ and $h$ are particularly interesting: as discovered by earlier works (Dordek et al., 2016; Sorscher et al., 2023), a choice of $h$ that imposes non-negativity constraint on the latent activities, such as ReLU, is necessary for the emergence of hexagonal grid cells. In our tPCN model, the activation functions are also important for biological plausibility: in both Equation 5 and Equation 6, the multiplication with the Jacobians $h'$ and $f'$ can be reduced to local, element-wise multiplications if $h$ and $f$ are element-wise nonlinearities such as ReLU and tanh. Although it is possible to design a circuit to perform the computations in softmax (Snow & Orchard, 2022), it is unclear how the Jacobian matrix of softmax can be computed in a biological circuit. Therefore, we first replace $f$ with a tanh function in our tPCN model and evaluate the model's performance in both path integration and its latent representations. As shown in Figure 5A, replacing $f$ with tanh results in slightly worse path integration performance and lower grid scores than the softmax baseline. However, visually, the latent representations are hexagonal and grid-like (Figure 5D), suggesting that using a biologically more plausible $f$ would not significantly affect the emergence of grid cells within tPCN. On the other hand, replacing the non-negative constraint (ReLU) on the latent activities with $h =$ tanh results in the amorphous latent representations (Figure 5E), which is consistent with Sorscher et al. (2023).

We next investigate the impact of the size of the environment, by training tPCN within a square environment of size 1.8m × 1.8m (big) and an environment of size 1.2m × 1.2m (small). Changing environment sizes does not affect the path integration performance, and does not affect tPCN's capability of developing grid cells either (Figure 5F for big environment and G for small environment). We also vary the number of latent neurons in the model from 256 to 512 and 1024, which does not affect the grid cell representations (Figure 5H shows the latent representations learned by a tPCN with 256 latent neurons). However, with fewer latent neurons, the performance in path integration becomes worse as the model has fewer number of parameters to perform the task (Figure 5A).

Earlier studies using PCNs to model visual representations have mostly used unsupervised PCNs (Rao & Ballard, 1999; Olshausen & Field, 1996; Millidge et al., 2024), which corresponds to blocking the velocity input $\mathbf{v}_t$ into tPCN in Figure 1B. Here we asked how removing velocity input would affect the path integration performance and grid cell emergence of tPCN. Mathematically, this is achieved simply by re-defining $\tilde{\mathbf{g}}_t := \mathbf{W}_r\hat{\mathbf{g}}_{t-1}$ without changing any inference or learning dynamics. It can be seen from Figure 5A that the path integration performance is significantly affected by the absence of velocity input, with an RMSE even higher than the stationary baseline, where the model does not predict any movement at all. Intriguingly, the latent representations developed by this unsupervised tPCN are still grid cell-like (Figure 5I) with a similar grid score distribution to the baseline model. This result demonstrates that grid cells can still emerge even in a model unable to perform path integration at all. Therefore, our model predicts that path integration is not a sufficient condition for the emergence of grid cells, which resonates with Schaeffer et al. (2022). In other words, it predicts that animals unable to navigate due to impaired velocity encoding may still develop grid cells as a result of self-motion.

## 5 DISCUSSION

**Relationship to Experimental Observations**  Here, we highlight properties of the biologically plausible circuit in Figure 1C, including those consistent with experimental observations, and those generating prediction about the hippocampal formation. This circuit can be naturally divided into a MEC layer and a hippocampal layer. The MEC layer contains velocity-encoding neurons ($\mathbf{v}$) and grid cells ($\mathbf{g}$), which aligns with experimental findings of the conjunctive representations of velocity and grids in the entorhinal cortex (Sargolini et al., 2006). In our model, grid cells in the MEC layer are recurrently connected through a specialized circuit involving interneurons $\hat{\mathbf{g}}_{t-1}$ that inhibit the output signal from the grid cells, allowing the error neurons $\boldsymbol{\epsilon}_t^{\mathbf{g}}$ to compute the temporal prediction errors. Experimental evidence suggests that lateral interactions in layer II of the MEC are predominantly inhibitory (Witter & Moser, 2006) and are mediated by interneurons such as basket cells (Jones & Bühl, 1993). Our model also predicts that these interneurons encode an eligibility trace $\hat{\mathbf{g}}_{t-1}$ from the immediate past. While recent studies have reported grid cells representing prospective locations (Ouchi & Fujisawa, 2024), it remains to be verified whether these cells are mechanistically supported by such "past" cells. Additionally, neurons in the entorhinal cortex are known to encode errors (Ku et al., 2021), suggesting a possible error-driven learning mechanism similar to that in tPCN.

In our model, the MEC and hippocampus are bidirectionally connected, a well-documented characteristic of entorhinal-hippocampal connectivity (Canto et al., 2008). Crucially, the circuit also posits the existence of error neurons $\boldsymbol{\epsilon}_t^{\mathbf{P}}$ in the hippocampus, which encode the discrepancy between place cell activities and inputs from MEC grid cells. The CA1 sub-region of the hippocampus has been shown to serve as a mismatch detector between the hippocampus and cortex (Lisman, 1999; Duncan et al., 2012). Our model predicts that in spatial navigation, the error neurons $\boldsymbol{\epsilon}_t^{\mathbf{P}}$ in the hippocampus, whose existence has been supported by Wirth et al. (2009) and Ku et al. (2021), can encode exactly this mismatch signal between the two regions.

**Conclusion**  In this work, we have demonstrated a biologically plausible learning rule for grid cells based on predictive coding. We have shown that with sparsity and non-negative constraints, classical PCNs can develop grid cell-like representations of batched place cell inputs. With inputs representing trajectories of moving agents, tPCN can also develop grid cell activations while performing path integration. We have developed a theoretical understanding of this property of tPCN by deriving and comparing its learning dynamics with that of BPTT, showing that unrolling a recurrent network is unnecessary for it to learn grid cells, and a more plausible approach with recursive inference dynamics should suffice. Furthermore, we have examined the robustness of our results by varying hyper-parameters of the model, and found that grid cells can be learned even without velocity inputs. Overall, our work demonstrates that predictive coding can serve as an effective and biologically plausible plasticity rule for neural networks to learn grid cells observed in the MEC. Importantly, compared with earlier learning rules specialized for grid cells, predictive coding is a general learning rule able to reproduce many other cortical functions and representations. Thus, our findings suggest that a single, unified plausible learning rule can be employed by the brain to find the most appropriate representation of cortical inputs in different regions.

## REPRODUCIBILITY STATEMENT

The code used for the experiments in this paper is provided as a zip file in the supplementary materials to facilitate reproducibility of our results. All hyperparameters for training are detailed in the appendix. Additionally, proofs for the theoretical results discussed in the paper are also included in the appendix for verification.

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

# A  APPENDIX

## A.1  ALGORITHMS

Below is the training algorithm for a sparse, non-negative PCN given spatial inputs $\mathbf{p}$. We obtain the grid cells shown in the main text directly by taking the converged latent activities $\mathbf{g}$ after training.

---

**Algorithm 1** Learning latent representations of space with a PCN

---

1: ▷ *Training*
2: **while** $\mathbf{W}$ not converged **do**
3:      Initialize $\mathbf{g}$ randomly;
4:      Input: $\mathbf{p}$
5:      **while** $\mathbf{g}$ not converged **do**
6:          $\mathbf{g} \leftarrow \texttt{ReLU}(\mathbf{g}_t + \Delta\mathbf{g}_t)$ (Eq. 2)
7:      **end while**
8:      Update $\mathbf{W}$ (Eqs. 3)
9: **end while**

---

Below is the training algorithm for tPCN in path integration tasks. The testing performance and grid cells shown in the main text are obtained by performing a forward pass through the model after training, given an unseen trajectory $\{\mathbf{v}_t, \mathbf{p}_t\}$.

---

**Algorithm 2** Path integration with tPCN

---

1: ▷ *Training*
2: **while** $\mathbf{W}_{\text{out}}, \mathbf{W}_{\text{r}}, \mathbf{W}_{\text{in}}$ not converged **do**
3:      Initialize $\hat{\mathbf{g}}_0$ randomly or from $\mathbf{p}_0$ via a PCN;
4:      **for** $t = 1, ..., T$ **do**
5:          Input: $\mathbf{p}_t, \hat{\mathbf{g}}_{t-1}$ and optionally $\mathbf{v}_t$
6:          Initialize $\mathbf{g}_t = f(\mathbf{W}_{\text{r}}\hat{\mathbf{g}}_{t-1})$
7:          **for** $k = 1, ..., K$ **do**
8:              $\mathbf{g}_t \leftarrow \mathbf{g}_t + \Delta\mathbf{g}_t$ (Eq. 5)
9:          **end for**
10:         Update $\mathbf{W}_{\text{out}}, \mathbf{W}_{\text{r}}, \mathbf{W}_{\text{in}}$ (Eqs. 6)
11:          $\hat{\mathbf{g}}_t \leftarrow \mathbf{g}_t$
12:      **end for**
13: **end while**

14: ▷ *Testing*
15: Initialize $\mathbf{g}_0$ randomly or from $\mathbf{p}_0$ via a PCN;
16: **for** $t = 1, ..., T$ **do**
17:      Input: $\mathbf{g}_{t-1}$ and optionally $\mathbf{v}_t$
18:      Obtain $\mathbf{g}_t, \hat{\mathbf{p}}_t$ via Eq. 7
19: **end for**

---

## A.2 DERIVATIONS OF LEARNING DYNAMICS

Here we derive the recurrent weight update rules for $\mathbf{W}_r^{RNN}$ (Equation 12) and $\mathbf{W}_r^{tPCN}$ (Equation 13). For RNN, we assume that the weights are updated at each time step and therefore $\mathbf{W}_r^{RNN}$ is updated following the chain rule:

$$\Delta \mathbf{W}_r^{RNN} = -\frac{d\mathcal{L}_{RNN,t}}{d\mathbf{W}_r} = -\frac{d\mathcal{L}_{RNN,t}}{d\mathbf{g}_t}\frac{d\mathbf{g}_t}{d\mathbf{W}_r} \tag{14}$$

We first look at the term $\frac{d\mathbf{g}_t}{d\mathbf{W}_r}$, which, following the rule of partial derivatives, can be written as:

$$\begin{aligned}
\frac{d\mathbf{g}_t}{d\mathbf{W}_r} &= \frac{\partial \mathbf{g}_t}{\partial \mathbf{W}_r} + \frac{\partial \mathbf{g}_t}{\partial \mathbf{g}_{t-1}}\frac{d\mathbf{g}_{t-1}}{d\mathbf{W}_r} \\
&= \frac{\partial \mathbf{g}_t}{\partial \mathbf{W}_r} + \frac{\partial \mathbf{g}_t}{\partial \mathbf{g}_{t-1}}\left(\frac{\partial \mathbf{g}_{t-1}}{\partial \mathbf{W}_r} + \frac{\partial \mathbf{g}_{t-1}}{\partial \mathbf{g}_{t-2}}\frac{d\mathbf{g}_{t-2}}{d\mathbf{W}_r}\right) \\
&= \dots \\
&= \sum_{k=1}^{t}\frac{\partial \mathbf{g}_t}{\partial \mathbf{g}_k}\frac{\partial \mathbf{g}_k}{\partial \mathbf{W}_r}
\end{aligned} \tag{15}$$

due to the recursive and implicit dependency of $\mathbf{g}_t$ on $\mathbf{g}_{t-1}$ and $\mathbf{g}_{t-1}$ on $\mathbf{W}_r^{RNN}$ for all $t$. Thus, the update rule can be written as:

$$\Delta \mathbf{W}_r^{RNN} = -\sum_{k=1}^{t}\frac{d\mathcal{L}_{RNN,t}}{d\mathbf{g}_t}\frac{\partial \mathbf{g}_t}{\partial \mathbf{g}_k}\frac{\partial \mathbf{g}_k}{\partial \mathbf{W}_r} \tag{16}$$

Since $\mathbf{g}_k = h(\tilde{\mathbf{g}}_k) = h(\mathbf{W}_r\mathbf{g}_{k-1} + \mathbf{W}_{in}\mathbf{v}_k)$, and $\mathcal{L}_{RNN,t} = \|\mathbf{p}_t - f(\mathbf{W}_{out}\mathbf{g}_t)\|_2^2$ the update rule can be written as:

$$\Delta \mathbf{W}_r^{RNN} = \sum_{k=1}^{t}\frac{\partial \mathbf{g}_t}{\partial \mathbf{g}_k}h'(\tilde{\mathbf{g}}_t)\mathbf{W}_{out}^\top f'(\mathbf{W}_{out}\mathbf{g}_t)\boldsymbol{\epsilon}_t^P \mathbf{g}_{k-1}^\top, \tag{17}$$

concluding our proof for Equation 12. The derivation for $\mathbf{W}_{in}^{RNN}$ is similar:

$$\Delta \mathbf{W}_{in}^{RNN} = -\frac{d\mathcal{L}_{RNN,t}}{d\mathbf{W}_{in}} = -\frac{d\mathcal{L}_{RNN,t}}{d\mathbf{g}_t}\frac{d\mathbf{g}_t}{d\mathbf{W}_{in}}, \tag{18}$$

and

$$\begin{aligned}
\frac{d\mathbf{g}_t}{d\mathbf{W}_{in}} &= \frac{\partial \mathbf{g}_t}{\partial \mathbf{W}_{in}} + \frac{\partial \mathbf{g}_t}{\partial \mathbf{g}_{t-1}}\frac{d\mathbf{g}_{t-1}}{d\mathbf{W}_{in}} \\
&= \frac{\partial \mathbf{g}_t}{\partial \mathbf{W}_{in}} + \frac{\partial \mathbf{g}_t}{\partial \mathbf{g}_{t-1}}\left(\frac{\partial \mathbf{g}_{t-1}}{\partial \mathbf{W}_{in}} + \frac{\partial \mathbf{g}_{t-1}}{\partial \mathbf{g}_{t-2}}\frac{d\mathbf{g}_{t-2}}{d\mathbf{W}_{in}}\right) \\
&= \dots \\
&= \sum_{k=1}^{t}\frac{\partial \mathbf{g}_t}{\partial \mathbf{g}_k}\frac{\partial \mathbf{g}_k}{\partial \mathbf{W}_{in}}
\end{aligned} \tag{19}$$

Therefore, the update rule for $\mathbf{W}_{in}$ in an RNN can be written as:

$$\Delta \mathbf{W}_{in}^{RNN} = \sum_{k=1}^{t}\frac{\partial \mathbf{g}_t}{\partial \mathbf{g}_k}h'(\tilde{\mathbf{g}}_t)\mathbf{W}_{out}^\top f'(\mathbf{W}_{out}\mathbf{g}_t)\boldsymbol{\epsilon}_t^P \mathbf{v}_k^\top \tag{20}$$

Finally, the update rule for $\mathbf{W}_{out}^{RNN}$ can be straightforwardly expressed as:

$$\begin{aligned}
\Delta \mathbf{W}_{out}^{RNN} &= -\frac{d\mathcal{L}_{RNN,t}}{d\mathbf{W}_{out}} \\
&= -\frac{d\mathcal{L}_{RNN,t}}{d\hat{\mathbf{p}}_t}\frac{d\hat{\mathbf{p}}_t}{d\mathbf{W}_{out}} \\
&= f'(\mathbf{W}_{out}\mathbf{g}_t)\boldsymbol{\epsilon}_t^P \mathbf{g}_t^\top
\end{aligned} \tag{21}$$

as there is no recursive dependency.

For tPCN, at each time step $t$ the following loss is minimized with respect to $\mathbf{W}_r$:

$$\mathcal{L}_{\text{tPCN},t} = \|\mathbf{p}_t - f(\mathbf{W}_{\text{out}}\mathbf{g}_t)\|_2^2 + \|\mathbf{g}_t - h(\mathbf{W}_r\hat{\mathbf{g}}_{t-1} + \mathbf{W}_{\text{in}}\mathbf{v}_t)\|_2^2 \tag{22}$$

Since $\hat{\mathbf{g}}_{t-1}$ is inferred through Equation 5, rather than forward-propagated by $\mathbf{W}_r$, the recursive dependency on $\mathbf{W}_r$ disappears, and thus the update rule for $\mathbf{W}_r$ can be locally derived as:

$$\Delta\mathbf{W}_r^{\text{tPCN}} = -\frac{d\mathcal{L}_{\text{tPCN},t}}{d\mathbf{W}_r} = h'(\tilde{\mathbf{g}}_t)\boldsymbol{\epsilon}_t^{\mathbf{g}}\hat{\mathbf{g}}_{t-1}^{\top} \tag{23}$$

If we also assume that the inference dynamics in Equation 5 have converged when the weights are updated, namely:

$$\Delta\mathbf{g}_t = 0 \Rightarrow \boldsymbol{\epsilon}_t^{\mathbf{g}} = \mathbf{W}_{\text{out}}^{\top}f'(\mathbf{W}_{\text{out}}\mathbf{g}_t)\boldsymbol{\epsilon}_t^{\mathbf{P}}, \tag{24}$$

the update rule can be written as:

$$\Delta\mathbf{W}_r^{\text{tPCN}} = h'(\tilde{\mathbf{g}}_t)\mathbf{W}_{\text{out}}^{\top}f'(\mathbf{W}_{\text{out}}\mathbf{g}_t)\boldsymbol{\epsilon}_t^{\mathbf{P}}\hat{\mathbf{g}}_{t-1}^{\top}, \tag{25}$$

which concludes our proof for Equation 13. Similarly, following the same assumption of converged inference and Equation 6, the update rule $\Delta\mathbf{W}_{\text{in}}^{\text{tPCN}}$ can be written as:

$$\Delta\mathbf{W}_{\text{in}}^{\text{tPCN}} = h'(\tilde{\mathbf{g}}_t)\mathbf{W}_{\text{out}}^{\top}f'(\mathbf{W}_{\text{out}}\mathbf{g}_t)\boldsymbol{\epsilon}_t^{\mathbf{P}}\mathbf{v}_t^{\top} \tag{26}$$

It can be seen that it differs from $\Delta\mathbf{W}_{\text{in}}^{\text{RNN}}$ only in the absence of the unrolling term $\frac{\partial\mathbf{g}_t}{\partial\mathbf{g}_k}$. On the other hand, the update rules $\Delta\mathbf{W}_{\text{out}}^{\text{RNN}}$ and $\Delta\mathbf{W}_{\text{out}}^{\text{tPCN}}$ are exactly the same.

### A.3 EXPERIMENTAL SETUPS AND HYPERPARAMETERS

**Place cell and trajectory parameters** We use DoS place cell encodings throughout most of our experiments. Formally, the activity of the $i$th place cell with this encoding, given a particular location $x$ can be written as:

$$K(x, C, \tau) := \exp\left(-\frac{(x - C)^2}{\tau \xi^2}\right) \tag{27}$$

$$p_i = \frac{K(x, C_i, 2)}{\sum_{j=1}^{N_p} K(x, C_j, 2)} - \frac{K(x, C_i, 4)}{\sum_{j=1}^{N_p} K(x, C_j, 4)} \tag{28}$$

where $C_i$ is the center of the place cell's firing field, and $\tau$ and $\xi$ define the width of the firing field's center and surround. The table below specifies parameters defining the place cells and trajectories:

| $\xi$ | Path length | Average agent speed | Environment size |
|---|---|---|---|
| 0.12m | 10 steps | $\{0.02, 0.05, 0.1\}$m/s | $\{1.4^2, 1.8^2, 2.0^2\}$m$^2$ |

Specifically, at time step $t = 0$, a 2D position and a head direction scalar in $[0, 2\pi]$ are randomly initialized. At each of the subsequent time steps, a random turn angle is sampled from a normal distribution and a random speed is sampled from a Rayleigh distribution. Both values are then multiplied by $dt$ mentioned in the main text. If the simulated agent hits a border wall at this time step, its speed is slowed and its turn angle is inverted. The position of the agent is updated according to the speed and turn angle at this time step. The trajectories are simulated using parameters adapted from the code provided in Sorscher et al. (2023).

**Model and training hyperparameters** In our experiments, we have used three models: sparsity and non-negativity constrained PCN, RNN and tPCN. The table below specifies parameters of model architectures:

| Model | $N_p$ | $N_g$ | $h$ | $f$ |
|---|---|---|---|---|
| sparse, non-neg. PCN | 512 | 256 | N/A | N/A |
| tPCN | 512 | $\{256, 512, 1024, 2048\}$ | $\{$ReLU, tanh$\}$ | $\{$softmax, tanh$\}$ |
| RNN | 512 | 2048 | ReLU | softmax |

The table below specifies hyperparameters used in training RNN and tPCN. We use `Adam` optimizer for all weight updates, and plain `SGD` for inference dynamics in tPCN. We found that in general, RNNs take more epochs to converge in the path integration task.

| Model | $N_x$ | batch size | learning rate | inference step size | epochs | inference iters | weight decay |
|---|---|---|---|---|---|---|---|
| tPCN | 50000 | 500 | $10^{-4}$ | $10^{-2}$ | 150 | 20 | $10^{-4}$ |
| RNN | 50000 | 500 | $10^{-4}$ | N/A | 200 | N/A | $10^{-4}$ |

The table below specifies hyperparameters used in training the sparse, non-negative PCN. We use `Adam` optimizer for all weight updates, and plain `SGD` for inference dynamics.

| $N_x$ | batch size | learning rate | inference step size | epochs | inference iters | weight decay | $\lambda$ |
|---|---|---|---|---|---|---|---|
| 900 | 100 | $2 \times 10^{-3}$ | $10^{-2}$ | 600 | 20 | $10^{-5}$ | 0.05 |

**Calculation of grid scores** The following grid score calculation process is adapted from Sargolini et al. (2006) and the code of Sorscher et al. (2023). It is summarized below for completeness and clarity:

- Get the rate map of latent neurons (potentially hexagonal grid cells);

- Place one copy of the rate map on top of the other, and start moving the top copy by $\delta \in \mathbb{R}^2$. If the rate maps are hexagonal grids, for particular $\delta$'s that make the firing peaks overlap, the autocorrelation between the stationary and moved maps will be 1; otherwise, the autocorrelation will be 0. We will then have a hexagonal autocorrelation map if the rate map itself is hexagonal;

- We then rotate the autocorrelation map and compute the correlation between each rotated map and the original map. If the rate maps are hexagonal, the correlation as a function of rotated degrees will be sinusoidal, with 60 and 120 degrees as peaks and 30, 90 and 150 degrees as troughs.

- Grid score is calculated as the minimum difference between the peak and trough correlation, which in theory is a real value in range $[-2, 2]$.

All experiments were performed on a single Tesla V100 GPU.

