# OpenReview forum: "Learning grid cells by predictive coding"
_ICLR.cc/2025/Conference — ICLR 2025 Conference Withdrawn Submission_

### Official Review · Reviewer_43mZ · 2024-10-28

**Soundness:** 4
**Presentation:** 4
**Contribution:** 3
**Rating:** 6
**Confidence:** 4

**Summary:**

The paper investigates the emergence of grid cells, known for their hexagonal firing patterns in spatial navigation, using predictive coding—a biologically plausible learning rule. The authors propose that grid cells can be learned by neural networks through predictive coding, which aligns well with the principles of local computations and Hebbian plasticity.

The key contributions are:

 - Demonstrating that predictive coding networks (PCNs) can naturally develop grid cell representations with sparse, non-negative constraints, and a temporal extension (tPCN) achieves similar results in dynamic tasks like path integration.
 - Establishing that tPCNs approximate the truncated backpropagation through time (BPTT), highlighting a biologically plausible alternative to BPTT for learning grid cells.
 - Analyzing the robustness of grid cell emergence in PCNs and tPCNs across varied architectural and environmental conditions, showing grid cells can still form even without velocity input.

**Strengths:**

**Originality:**
This paper provides a new perspective on grid cell formation by applying predictive coding. While previous work has used RNNs trained with BPTT to simulate grid cells, this study introduces predictive coding networks (PCNs) and temporal PCNs (tPCNs) as biologically plausible alternatives. While predictive coding has been addressed in hippocampal formation previously (Stachenfeld et al. ++), the proposed learning rules are novel in this context.

**Quality:**
The authors demonstrate grid cell emergence in PCNs and perform a comparative analysis with existing RNN models. By analytically showing that tPCNs approximate truncated BPTT, they provide a theoretical solid grounding for their approach. Further, the robustness analysis—exploring different model architectures, non-linearities, and environments—addresses shortcomings proclaimed in recent work (Sorscher vs. Schaeffer). The theoretical and empirical sections are well-integrated.

**Clarity:**
The authors use clear visual representations of presented ideas, making interpretation intuitive.  The derivations are well-presented, especially in demonstrating the correspondence between tPCNs and truncated BPTT. However, some technical details on the inference dynamics of tPCNs might benefit from additional clarity or simplification, especially for readers less familiar with predictive coding.

**Significance:**
The findings are interesting for neuroscience and machine learning. They suggest that predictive coding may underpin not only perceptual but also spatial and navigational representations. For neuroscience, predictive coding may unify perspectives across cortical functions. For machine learning, it offers an alternative to backpropagation-based learning in dynamic systems.

**Weaknesses:**

Although it is nice to see grid cells emerge in the proposed setup, it is not that surprising given the setup with static place cell readout. The comparison between BPTT and tPCNs is more interesting, in my opinion, than the grid cell results and can have broader implications beyond this particular setting; I would present this as the main result and, therefore, consider moving this result to an earlier stage and presenting the grid cell stuff as a test case.

The model operates under certain assumptions (e.g., reliance on sparsity, non-negative constraints, simplified path integration tasks, and place cell readout) that may not generalize well across different types of neuronal representations or tasks. However, the discussion lacks a critical assessment of these assumptions, specifically regarding where the predictive coding model might fall short compared to other frameworks for grid cells, such as the recent development of self-supervised learning for grid cells ([Schaeffer et al.](https://arxiv.org/abs/2311.02316)), conformal isometry, or distance preservation ([Xu et al.](https://arxiv.org/abs/2210.02684), [Dorell et al.](https://arxiv.org/abs/2209.15563)). For example, the choice of static read-out place cells limits studies of remapping (but can be done; see [Schøyen et al.](https://www.sciencedirect.com/science/article/pii/S258900422302179X), different geometries [Krupic et al.](https://www.nature.com/articles/nature14153) etc.

The proposed predictive coding model successfully generates grid cells, but the mechanistic explanation for how and why grid cells emerge under predictive coding is lacking. Moreover, the field suffers from challenges in comparing representations across studies, barring visual inspection. Grid scores are used to assess grid cell likeness; however, these give little insight beyond 60-degree symmetry. I suggest you use something else to assess the function of the networks, such as ablation studies and studying the full representational setting of the network. For example, do you see border cells, band cells, etc? At least provide examples, preferably representations from the full network, in the supplementary.

All in all, since the title and introduction of the paper highlight grid cells, I would expect more analysis of this finding and a broader comparison with the existing literature. However, I think the more interesting finding is the comparison between BPTT and tPCNs. Therefore, I would recommend lifting this part of the paper and proposing the grid cell story as a potential application motivating further studies on this line of work, although I do see your point on extended analysis on this being out of scope.

**Questions:**

The authors find that grid cells emerge under various configurations and constraints, even in the absence of velocity input. Could they expand on the implications of this finding for the role of predictive coding in spatial learning?

You claim that tPCN approximates tBPTT; however, the RMSE indicates that when the inference has fully converged, the tPCN outperformed tBPTT. Path integration is a Markov process, and it therefore makes sense that tBPTT should work. However, as you show, having the extra inference steps helps. Is it then tPCN that approximates tBPTT or the other way around (tBPTT approximates tPCN)

Moreover, this begs the question: what is the difference between $g_{t-1}$ from RNNs and $\hat{g}_{t-1}$ tPCNs that give this performance boost?

Is there a qualitative difference in grid cells between the models, or are there other cell types that make $\hat{g}_{t-1}$ "better"? One way to hint at this would be to ablate neurons in $g$ and rank them according to their effect on the loss. Are there any differences between these two populations? Another way would be to perform a detailed analysis of the predictive power of $g$ cells in the two models, for example, according to Ouchi et al.

Related works, such as the work from Giocomo in 2011, are outdated. Whether oscillatory dynamics are important for grid cells started as you point out with the work by [Burgess](https://pmc.ncbi.nlm.nih.gov/articles/PMC2678278/) and Hasselmo, but it was later included in CANNs by [Bush and Burgess](https://pubmed.ncbi.nlm.nih.gov/24695724/). The importance of oscillations in grid cells has been tested experimentally by  [Lepperød et al](https://www.science.org/doi/full/10.1126/sciadv.abd5684), [Schmidt-Hieber et al.](https://www.nature.com/articles/nn.3340), [Robinson et al.](https://www.sciencedirect.com/science/article/pii/S2211124724009197)

**Minor**

 - $\hat{g}$ is used but not introduced as inferred before line 392; this can be nice to point out earlier.
 - Whether grid cells are learned or are there from birth is disputed; I would present this in less certain terms.

---

> ### Author Response · Authors · 2024-11-24
>
> I would like to express my sincere gratitude for your thoughtful and insightful comments on our paper. Your feedback on improving the depth of exploration into both the grid cell emergence in tPCN and the algorithmic comparison to BPTT is very important to us. While we have decided to withdraw the submission at this stage, your suggestions will play a key role in guiding the revisions for future submission. Thank you again for your time and effort in providing such constructive feedback.

---

### Official Review · Reviewer_L2F6 · 2024-11-01

**Soundness:** 2
**Presentation:** 2
**Contribution:** 1
**Rating:** 3
**Confidence:** 3

**Summary:**

This study demonstrates that predictive coding can effectively train neural networks to develop hexagonal grid representations from spatial inputs, providing a biologically plausible explanation for the emergence of grid cells in the medial entorhinal cortex. By analytically comparing predictive coding with existing models, we offer new insights into the learning mechanisms of grid cells and extend predictive coding theory to the hippocampal formation, suggesting a unified learning algorithm for various cortical representations.

**Strengths:**

The paper is clearly written, and the question is well-defined.

**Weaknesses:**

My major concern is that the work may lack novelty.

1. The use of non-negative and sparse network designs to produce grid cell-like patterns has been extensively discussed. For example, [1] reported that non-negative and sparse properties can generate grid cell -like patterns and theoretically demonstrated why non-negativity is the main driver of grid cell formation (which the author's  paper does not address) instead of sparsity. Similar findings were also reported in [2]. Earlier, [3] proves that a nonnegativity constraint on firing rates induces a symmetry-breaking mechanism which favors hexagonal firing fields. [4] further explored, through extensive experiments, the conditions necessary for generating grid cells.

2. Prediction tasks, including path integration, that produce grid cell-like patterns have also been widely reported, especially when the input data takes a place cell-like form. For instance, [5] also used place cell like input and path integration tasks to train networks and generate grid cells, while [6] theoretically analyzed the role of predictive learning in forming low-dimensional representations.

3. In my understanding, tPCN is very similar to a one-step RNN (apart from the difference in local learning rules), so the fact that its training process resembles that of one-step tBPTT is not surprising. As previously noted, the key to forming grid cells lies in the predictive task, not the RNN network itself. Therefore, the similarity between tPCN and RNN does not offer significant insight into the generation of grid cells.

For the reasons above, I believe this paper does not offer substantial novelty or make a clear contribution to the field.



[1]Whittington, James CR, et al. "Disentanglement with biological constraints: A theory of functional cell types." *arXiv preprint arXiv:2210.01768* (2022).

[2]Dorrell, William, et al. "Actionable neural representations: Grid cells from minimal constraints." *arXiv preprint arXiv:2209.15563* (2022).

[3]Sorscher, Ben, et al. "A unified theory for the origin of grid cells through the lens of pattern formation." *Advances in neural information processing systems* 32 (2019).

[4]Schaeffer, Rylan, Mikail Khona, and Ila Fiete. "No free lunch from deep learning in neuroscience: A case study through models of the entorhinal-hippocampal circuit." *Advances in neural information processing systems* 35 (2022): 16052-16067.

[5]Whittington, James CR, et al. "The Tolman-Eichenbaum machine: unifying space and relational memory through generalization in the hippocampal formation." *Cell* 183.5 (2020): 1249-1263.

[6]Recanatesi, Stefano, et al. "Predictive learning as a network mechanism for extracting low-dimensional latent space representations." *Nature communications* 12.1 (2021): 1417.

**Questions:**

see weakness

---

### Official Review · Reviewer_sDso · 2024-11-02

**Soundness:** 2
**Presentation:** 3
**Contribution:** 2
**Rating:** 3
**Confidence:** 3

**Summary:**

This paper proposes that the mechanism by which grid cells are learned in biological systems may involve predictive coding. To test this hypothesis, the authors trained both a predictive coding network (PCN) and a temporal predictive coding network (tPCN) on path integration and non-path integration tasks. They observed that hexagonal firing patterns, characteristic of grid cells, emerged in both paradigms. Since PCN and tPCN introduce error cells that enable learning with spatially and temporally local rules, this discovery suggests a biologically plausible mechanism for grid cell formation. The authors also analyze the learning process in tPCN, comparing it analytically with 1-step backpropagation through time (BPTT), to explain the emergence of grid cells. Finally, they assess the robustness of grid cell emergence in their model by testing various activation functions, environments, and network sizes.

**Strengths:**

To the best of my knowledge, this paper is the first to suggest that a predictive coding network can serve as a biologically plausible model for learning grid cells and perform simulations to validate this hypothesis. Additionally, the paper extends the application of PCN’s locally-based learning method to approximate backpropagation (BP) in temporally processing networks, using tPCN. While not formally proven, the authors draw comparisons between tPCN and 1-step BPTT, indicating that with multi-step inferences, tPCN’s performance could approach that of BPTT.

**Weaknesses:**

The main limitation lies in novelty. First, previous studies have already shown that grid cells can be learned either through non-negative PCA or via a single-layer BP-based network from place cell activity. Likewise, RNNs trained via BPTT for path integration to predict place cell activity have also been reported (see Sorscher et al., 2022). Additionally, the ability of PCN to approximate BP using local learning rules has been demonstrated previously (see Song et al., 2020), and the t-PCN structure’s capacity to approximate BPTT is a straightforward extension of prior work (Millidge et al., 2024). The robustness analysis in this paper largely follows procedures established in earlier RNN studies and does not report new phenomena (Schaeffer et al., 2022). Other biologically plausible learning algorithms, such as those using Oja’s rule, have also achieved grid cell-like activity, suggesting that this paper’s algorithm is not unique in this regard. Overall, the contribution seems to synthesize existing ideas without introducing significant innovation.

**Questions:**

I have two questions:

1. In both model architectures presented, grid cell activity depends on input from place cells. However, in biological systems, place cell activity varies significantly across different environments, showing a phenomenon known as global remapping, whereas grid cells maintain a stable 2D toroidal manifold across environments. How does this model account for this discrepancy? If place cell activity, the input source for grid cells, changes substantially across environments, how does the model explain the stability of grid cell activity?

2. In the medial entorhinal cortex (MEC), grid cells are organized into modules with distinct spacings. In the model proposed in this paper, do the network’s grid cells display discrete spacing distributions, and are there any indications of modular independence in their connectivity?

---

### Official Review · Reviewer_aP8i · 2024-11-04

**Soundness:** 3
**Presentation:** 3
**Contribution:** 1
**Rating:** 5
**Confidence:** 4

**Summary:**

The authors investigate how a temporally-dependent version of predictive coding can extract compact latent spaces in the form of periodic grid activity from temporally structured place cell input. The findings are of general interest to theories of learning in biological settings, and replicate many previous results with a more biologically plausible learning mechanism.

**Strengths:**

-	The general finding of the tPCN is encouraging, and the generalization to a different task than the Millidge 2024 paper is promising.
-	The robustness experiments (4.4) show that the emergent grid-like activity is robust to model architectures. This is encouraging, since many experimental neuroscience manipulations show grid cells to be robust to manipulations of the environment of neural activity.

**Weaknesses:**

-	Overall, the study seems like an incremental follow on of the tPCN paper applied to a new domain, but which does not require fundamental changes to the original algorithm.
-	The path integrating tPCN assumes input in the form of place cell activity, but does not account for how place cells and grid cells form from the combination of visual and self-motion information. Combined with the lack of anatomical constraints of direction of connectivity, the study is more about the formation of compressed latent spaces than the medial temporal lobe. Several existing studies, largely cited in the paper, already investigate the formation of such successor representations by predictive coding.
-	The authors dismiss previous examples of learned grid cells (Dordek, Stachenfeld, Schaffer, etc) on the basis that these are not biologically plausible learning methods, but then move to use real-valued activation functions. There is no evidence from the methods presented in this paper that a spike-based temporal predictive coding network would converge.

**Questions:**

- The soundness of the paper is high, but my primary concerns center around the novelty of the algorithm beyond tPCN itself. Simply applying a non-negative constraint and applying to a new task does not seem like a sufficiently novel contribution for ICLR. It is unclear what enhancements of the algorithm could be necessary in the context of spatial navigation.

---

### Note · Authors · 2024-11-24

I have read and agree with the venue's withdrawal policy on behalf of myself and my co-authors.